# LEARNING SEMANTICS, NOT ADDRESSES: RUNTIME NEURAL PREFETCHING FOR FAR MEMORY

## ABSTRACT

Memory prefetching has long boosted CPU caches and is increasingly vital for far-memory systems, where large portions of memory are offloaded to cheaper, remote tiers. While effective prefetching requires accurate prediction of future accesses, prior ML approaches have been limited to simulation or small-scale hardware. We introduce FarSight, the first Linux-based far-memory system to leverage deep learning by decoupling application semantics from runtime memory layout. This separation enables offline-trained models to predict access patterns over a compact ordinal vocabulary, which are resolved at runtime through lightweight mappings. Across four data-intensive workloads, FarSight delivers up to 3.6× higher performance than the state-of-the-art.

## 1 INTRODUCTION

In response to increasing application memory demands and the slowing down of server main memory scaling, major datacenters like Google and Microsoft have adopted the datacenter architecture of *far memory*, where most of the data resides in remote, network-attached memory and only a small subset is cached locally in CPU-attached DRAM or GPU HBM (Chen et al., 2023; Lu et al., 2024; Liu et al., 2025). Far-memory systems offer lower cost/GB, higher energy efficiency, lower carbon footprint, and elastic capacity. However, the main limitation of today's far-memory systems is the application performance overhead caused by accessing non-local data from remote memory, an operation that is often more than 20 times slower than a local DRAM access. A common mechanism employed in existing far-memory systems is to fetch data from far memory in the background before they are accessed (*i.e.*, *prefetching*), thereby avoiding the foreground, *on-demand* far-memory data fetching time that directly affects application performance.

Effective prefetching relies on the accurate prediction of future memory accesses. Today's far-memory systems take a conservative approach of only prefetching accesses that follow simple rules (*e.g.*, sequential or strided access patterns (Guo et al., 2023; Maruf & Chowdhury, 2020)). However, most data-center workloads, such as graph processing (Lawrence et al., 1998; Han et al., 2024), tree and index structures (Guttman, 1984; Gusfield, 1997), pointer chasing (Hsieh et al., 2017), and recursive data structures (Harold & Means, 2004), exhibit memory access patterns that defy rule-based prefetching. As such, they still suffer from huge performance overheads, preventing the wide adoption of far-memory systems and their potential cost saving which datacenters could have otherwise achieved.

We propose *FarSight*, a deep-learning-based prefetching mechanism designed for far-memory systems and implemented in the Linux kernel. Different from simulation- or hardware-based CPU cache prefetchers, a key challenge in software-based, far-memory prefetchers is to avoid prediction latency overhead while achieving high accuracy in a vast search space. For example, any online training or even traveling through the PCIe bus to a GPU for model inference would significantly slow down application execution. Representing memory accesses directly using their addresses in a model is also infeasible (a 64-bit machine has $2^{48}$ virtual memory addresses).

Our insight is that memory access behavior is governed by both application semantics (*e.g.*, algorithmic logic) and input-dependent runtime context (*e.g.*, memory layout). The semantics tend to generalize across inputs and can be learned offline, but the actual memory addresses are input-specific and best handled at runtime. We exploit this separation by training a deep-learning (DL) model to learn semantic patterns and delegating address resolution to a runtime system component.

Specifically, we propose to represent application semantics as relationships between memory accesses. For each access, we observe that the subsequent access usually only has a small set of possibilities. We assign each possible outcome an *ordinal*. While the actual memory addresses corresponding to these possibilities vary across different inputs, the transition pattern—*i.e.*, which ordinal is likely to follow given the history——is often learnable and generalizable. For instance, in a linked list traversal, each accessed node is always followed by the next node in the list. Although the memory addresses of these nodes differ per execution, the access behavior remains the same.

We set the DL model vocabulary as the anticipated outcome possibilities (*i.e.*, a configurable $K$ defaulting to 64). The DL model uses memory access history sequences encoded in the vocabulary $K$ and predicts future memory accesses as a sequence of ordinals ranging from 0 to $K - 1$. These ordinals are resolved at runtime via a lightweight *future map*——an in-memory mapping table we propose to record for each accessed memory. A future map contains $K$ entries, each mapping (from its index) to a memory address observed at runtime to follow the access to the current page. This design significantly reduces model vocabulary from the full memory address space to a small, fixed size, enabling high prediction accuracy with a compact DL model.

Our model builds on top of Retentive Network (Sun et al., 2023), a compact Transformer-variant model small enough to fit within a CPU core's L1 cache. Apart from the above representation, we propose several techniques to further reduce or hide the model prediction and prefetching latencies. We use a position encoding scheme that supports reuse of cached context across predictions, reducing redundant computation and memory access overhead. We predict several accesses ahead so that prefetched data can arrive at the local memory before being accessed, and we perform the prediction and prefetching in the background to hide their latencies.

We implement FarSight's training as an offline process when an application is deployed. We implement FarSight's runtime system in the Linux kernel, with most of it being a Linux kernel module and the rest changing Linux's swap system slightly. We evaluate FarSight with four real-world applications and benchmarks: MCF on SPEC 2006 Benchmark (Henning, 2006), PageRank and Shortest Path from the GAP benchmark suite (Beamer et al., 2015), and XGBoost (Chen & Guestrin, 2016). We compare FarSight to a SoTA DL-based memory prefetcher, Twilight (Duong et al., 2024), and two non-ML-based far memory systems, FastSwap (Amaro et al., 2020), Hermit (Qiao et al., 2023). Our results show that FarSight outperforms FastSwap by up to 3.6 times, Hermit by up to 2.6 times, and Twilight by up to 3.9 times.

## 2 BACKGROUND, RELATED WORKS, AND MOTIVATION

### 2.1 FAR MEMORY AND EXISTING PREFETCH APPROACHES

Far-memory systems are systems where applications have access to memory beyond CPU-local memory, *e.g.*, memory at another server or memory in a disaggregated memory pool. Far memory allows applications to access larger amounts of cheaper (*e.g.*, older generations of, non-volatile, pooled) memory. In practice, to strive for higher memory resource efficiency, local memory sizes are often set to below half of the far memory size (Chen et al., 2023). For applications to utilize far memory, there usually is an indirection layer (*e.g.*, a swap system (Amaro et al., 2020; Qiao et al., 2023) or a user-space library (Ruan et al., 2020; Guo et al., 2023)) that fetches data from far to local memory.

The main limitation of far-memory systems is the communication delay between local and far memory. For example, for a local memory size that is half of the far memory size, naive implementation of a far-memory system could result in half of the accesses going to far memory, resulting in an application slowdown of *13 times* for RDMA-based far-memory systems. To hide this delay, most far-memory systems prefetch future accesses from far memory and cache them locally. Existing far-memory systems (Amaro et al., 2020; Ruan et al., 2020; Maruf & Chowdhury, 2020) prefetch far-memory data with rule-based approaches by detecting and following linear and strided patterns. As such, they are limited to only benefit applications with such regular memory access patterns.

Prior research works (Shi et al., 2021; Hashemi et al., 2018; Srivastava et al., 2019; Peled et al., 2015; 2019) explored using ML techniques for local server prefetch predictions at the micro-architecture level by prefetching data from memory into CPU cache. However, because of their performance

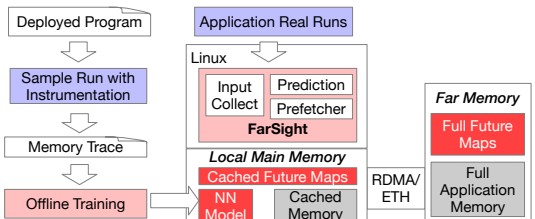

Figure 1: FarSight overall architecture *all red parts are FarSight.*

and/or accuracy issues, they have only been realized in simulation or for offline trace analysis. For example, (Peled et al., 2015) proposed reinforcement-learning-based and regression-based (Peled et al., 2019) approaches for memory prefetching. The former sets up the prefetching prediction as a classification problem and can only accommodate *four* possible address offsets for each prediction. The latter has accuracy issues as a regression model aims to be close to the ground truth, but correct prefetch requires the exact truth. Twilight and T-LITE (Duong et al., 2024) use the combination of a customized two-layer neural-network model, clustering, and frequency-based history table for CPU cache prefetching. DART (Zhang et al., 2023) distills a transformer model and then transforms the distilled model into a hierarchy of table lookups to reduce runtime performance overhead. Although these works have shown their CPU cache prefetch effectiveness through simulation, their training and prediction processes are complex and lack generalization or consistent accuracy.

### 2.2 CHALLENGES OF DL FOR FAR MEMORY PREFETCHING

Successful application of DL for far memory prefetching presents several unique challenges.

**Latency.** Prefetches are on the performance-critical path, directly impacting application performance, and prefetched data that arrives later than when an application accesses it is useless. To put things in perspective, it takes close to $10\mu s$ to launch a GPU kernel of just *one* matrix multiplication and a CPU memory copying kernel, while a local DRAM access takes less than $1\mu s$ and a far-memory 4 KB page read takes around $2\mu s$ with today's InfiniBand-based network. Pausing an application for $10\mu s$ to perform a prediction is unacceptable, and by the time a prediction finishes, hundreds of thousands of memory accesses could have happened, making the prefetched data stale.

**Accuracy.** Although wrong prefetches do not affect application execution correctness, they waste local memory space and network bandwidth, which are especially precious under far-memory environments. As local memory is expected to run at full capacity, a prefetched page will need another local page to be swapped out on demand, taking about $4\mu s$ from our evaluation. Thus, it is essential to design ML techniques for accuracy.

**Generalization.** Because of the performance requirement, it is infeasible to train or fine tune a model at run time. An offline-trained model avoids any runtime overhead but has no access to runtime status. An application's memory accesses at runtime are often input-dependent. Moreover, memory addresses can be different across runs even with the same input because of memory address randomization techniques like address space layout randomization (ASLR).

To make DL-based prefetch practical for far-memory systems, latency, accuracy, and generalization must be achieved *simultaneously*.

## 3 FARSIGHT DESIGN

FarSight consists of an offline training component and an online predictor and prefetcher component sitting in the Linux kernel's swap system, as illustrated in Figure 1. When an application is deployed, FarSight trains a small model (3K-parameter Retentive Network (Sun et al., 2023) architecture) by tracking the execution of the application with user-supplied sample application inputs. During runtime, FarSight loads the trained model onto each CPU core running the application. FarSight predicts future far-memory accesses using its captured recent program execution history and issues the corresponding prefetch requests.

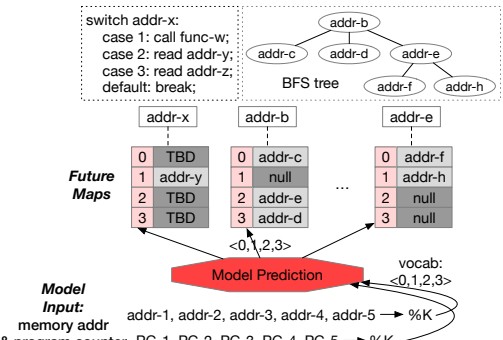

Figure 2: FarSight prediction representation *An example of vocabulary size (K) being 4. The top part shows code/algorithm corresponding to the accesses of chunks* `addr-x`, `addr-b`, *and* `addr-e`. *The bottom shows the input to the model: the chunk addresses and PCs of the 5 previous misses.*

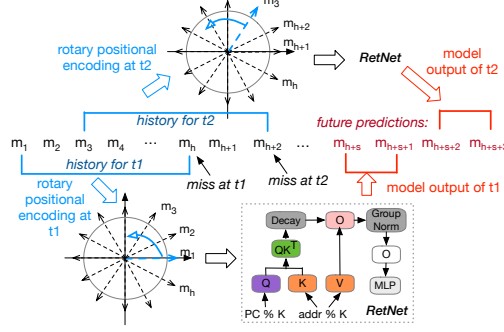

Figure 3: FarSight's prediction optimization methods *Demonstrating the use of each history window to predict $s$ misses ahead of time and predicting $f = 2$ pages at a time.*

## 3.1 PREDICTION TASK FORMATION

To reduce runtime overhead and improve the accuracy of a small DL model, FarSight uses a small vocabulary size of $K$, defaulted to 64. Below, we explain FarSight's prediction process and how we achieve this small vocabulary size with our pattern-addressing decoupling idea.

**Model inputs.** FarSight uses page miss history as the input to the DL model instead of full memory access history. This is because, by being in the swap system, FarSight can observe and log miss addresses on every page fault without incurring additional overhead. In contrast, capturing the full memory access stream would introduce substantial runtime overhead and is therefore avoided. In addition to using page miss addresses, we associate every miss with the faulting program counter (PC), as doing so can incorporate program execution information with memory access history, and recording and using PC incurs no additional overhead.

To fit the two types of inputs into the vocabulary, we take the mod of their value to the vocabulary size, $K$. We then use a history sequence of $h$ pairs of the modulo of miss page address and PCs as the model input, as shown at the bottom of Figure 2. Although taking a mod is a lossy process, a history sequence and two types of information allow our model to make accurate predictions.

**Model outputs and future maps.** We choose to predict page misses (*i.e.*, accesses to memory pages not in local memory), rather than attempting to predict every individual memory access——which would be computationally intensive and unnecessary. Essentially, FarSight uses page miss sequence in recent history to predict page miss sequence in the future. This approach significantly reduces the computational load on the DL model and the monitoring overhead.

A straightforward way to model memory miss prediction is by using their memory addresses, as used by most prior ML-based memory access prediction works (Shi et al., 2021; Zhang et al., 2023; Hashemi et al., 2018; Peled et al., 2015; 2019). While straightforward, address-based prediction requires a huge vocabulary—$2^{36}$ for 4 KB pages on 64-bit machines. In comparison, English vocabulary used by modern LLMs is only 50K to 100K in size (Radford et al., 2019; Sugimoto, 2023), beyond which prediction accuracy starts to degrade even for large models. Clearly, the huge memory address vocabulary does not meet far-memory prefetching's accuracy demands (§2).

Our solution is to label possible outcomes of memory access as *ordinals*. Specifically, we record a vocabulary size (*i.e.*, $K$) of possible next memory page misses after a miss happens at page $X$. Based on the model inputs as described above, our model predicts an ordinal from 0 to $K - 1$, corresponding to one of the likely next page misses. We dynamically maintain a *future map* for each page $X$ in the *local memory*. Each entry in the future map represents one possible page to be accessed after the miss of page $X$. When a page $Y$ is accessed after $X$ and our predicted ordinal is $k$, we fill the $k$th entry in $X$'s future map with the virtual memory address of page $Y$. A null future map entry represents an outcome that has not yet occured during runtime.

Figure 2 illustrates this idea with two example code patterns. Each page is associated with its future map of size 4 (*i.e.*, a vocabulary size of $K = 4$). As an example, `addr-b` is a tree node that has previously been followed by accesses of `addr-c`, `addr-e`, and `addr-d`. Thus, the memory addresses for memory pages containing `addr-c`, `addr-e`, and `addr-d` have been recorded in the future map of `addr-b`. When `addr-b` is accessed again, the model uses memory access and PC history to predict one of $< 0, 1, 2, 3 >$. If 1 is predicted, no prefetch will be performed. Otherwise, the corredponding memory page will be prefetched.

**Vocabulary size.** Naturally, a program can have fewer or more than $K$ possible memory pages to access after a page is accessed. If there are fewer possibilities (*e.g.*, pages `addr-b` and `addr-e` only have three and two possible outcomes), some of the future map entries will not be used, wasting local memory. If there are more possibilities than $K$, the model will not properly capture the less frequently occurring accesses.

We set the configurable $K$ to 64 by default, which strikes a balance of memory overhead and prediction accuracy. Note that $K$ outcomes represent $K$ 4 KB pages, which contain a $4K$ KB address range (256 KB by default). The default value of 64 works well for all our applications for a few reasons. First, small future maps allow for hot entries to be cached at CPU L1 and L2 caches, largely reducing the prediction latency. Second, applications with repeatable behavior usually have limited possible outcomes (subsequent accesses) after one page fault. For example, pointer chasing, database B-trees, common program control flows, and sorting algorithms have one to a handful of possible memory outcomes. On the other hand, a graph with high skew could have some nodes with a large number of neighbors, but the frequency of accessing these neighbors is relatively low, and failure of prefetching them does not impact application performance much, as shown by our PageRank results (§4.1). Third, most allocators assign addresses from a range to closely requested address allocations, resulting in most accesses being within the same memory page and $K$ pages being able to host them.

**DL model architecture.** We adopt the Retentive Network (RetNet) model architecture (Sun et al., 2023), which unifies the benefits of Transformer (Vaswani et al., 2017) and RNN (Socher et al., 2011). It replaces the Transformer's softmax operation with a weighted (giving more weight to more recent history) sum of the sequence's history context, as shown in Figure 3. We feed the memory address (after mod $K$) as Q and PC mod $K$ as K and V.

The assumption that recent tokens are more important than distant ones may not hold for natural language prediction. However, it fits memory prediction well, as program behaviors are usually influenced more by recent history than distant history.

RetNet achieves O(1) inference latency and O(N) inference memory space, where N is the sequence length, while maintaining good accuracy and training speed. Its superior inference latency and memory consumption allow us to deploy it on each CPU core. In contrast, other sequence-to-sequence models such as vanilla Transformer and RNN do not fit our latency and memory needs, since their inference either grows linearly with sequence length or involves costly trigonometric computations. State Space Models (SSMs) are another class of architectures offering constant-time inference and relying primarily on matrix operations. Importantly, RetNet, which we adopt, belongs to this family, as (Gu & Dao, 2023) highlights that it is a special case of a linear SSM.

## 3.2 PREDICTION METHOD OPTIMIZATION

Based on our basic prediction process as discussed above, we propose a set pf optimization methods to further improve FarSight's overall performance.

**Look-ahead, batched prefetch.** So far, we assume the model predicts the immediate next missed page. With such an approach, even if we issue a far-memory access request right after the prediction, the communication delay is likely longer than when the next miss happens, making the prefetched data arrive too late to be useful. Our solution is to predict farther into the future with a *look-ahead distance* to cover the communication delay to prefetch application data. Specifically, we predict and prefetch the $s$th future memory miss from the current access (*i.e.*, $s$ is the look-ahead distance). We determine $s$ by conservatively choosing a large percentile (*e.g.*, 95%) of profiled communication delay distribution, $d$, and the average profiled inter-arrival time between two memory accesses, $l$; $s$

is $d/l$. With this conservative setup, prefetched data could arrive before it is needed but rarely after. Furthermore, to efficiently utilizing network bandwidth, we prefetch $f$ pages at a time.

**Model input encoding.** At the time of a miss, we use the recent history window of $h$ misses as the model input sequence. A naive way to encode the history is to treat each miss as one token and perform positional encoding of these tokens starting from position 0. This encoding works well for generic sequence-to-sequence problems, as each new request to the model is treated as a different sequence. However, in our environment, most tokens (past accesses) but one (the most recent access) overlap when we move the history window by one position ($m_3$ to $m_h$ accesses overlap between the two predictions shown in Figure 3). With the naive encoding method, we would need to recompute everything since the token positions have changed in the new window.

To solve this problem and improve prediction performance, we propose a new encoding method based on rotary positional encoding (Su et al., 2021). Instead of always starting from 0, our encoding starts from the position where the first access in the current history window (*e.g.*, $m_3$ in t2 window) was in the previous window ($m_3$ at the 60-degree angle). Essentially, we turn the rotary wheel by one unit of angle at every prediction step to align the same access at the same angles. This allows us to reuse the computed context of overlapped accesses (*e.g.*, $m_3$ to $m_h$).

**CPU-based background model prediction.** To minimize the impact on application performance, we try to hide FarSight's prediction time behind foreground application execution. We perform prediction at the CPU core of each application thread while the thread waits for on-demand far-memory I/O operations. We maintain all model weights, application memory access history (model inputs), and hot future map entries in the CPU core's L1 and L2 caches. This is feasible because of the small model size and small history window we choose (totalling 20 KB of weights and metadata). As such, one model prediction takes less than 600ns, significantly faster than a network round trip of around two microseconds with RDMA. Furthermore, we perform all prefetching asynchronously.

### 3.3 MODEL TRAINING AND INPUT GENERALIZATION

Each deployed application goes through an offline training process once until its usage pattern significantly shifts. This per-application training practice is acceptable in our targeted *on-premise* data-centers where applications are used for enterprise only. In such environments, there are typically tens to hundreds of applications per data center, and huge application input drift is rare (Chen et al., 2016; Panchenko et al., 2019).

To train a model, we execute the application with a user-supplied sample input with fully local memory on a single server. We expect the sample input to be smaller than the actual runtime inputs and can thus run fully locally. As will be shown in §4.1, a model trained with small inputs generalizes well to different larger inputs thanks to FarSight's decoupled representation. We instrument the sample run to capture memory accesses and then train the RetNet model with this collected trace in an offline manner. The training process uses the same vocabulary size, look-ahead distance, encoding method, and history length as introduced in §3.1 and §3.2. As we have the oracle knowledge of the whole execution, we maintain the top $K$ most frequently accessed subsequent pages in each future map. The training target is the correct index in the future map that matches the ground truth.

Now, let us understand why a small training input could generalize to various larger inputs during run time. FarSight makes its prediction based on "paths" of memory access "branches", essentially how a program "branches" next based on runtime history behavior. For paths visited during training, the model is likely to correctly predict the next step if the path is visited again at run time. For example, when a program has a sequence of memory accesses with no conditional branches, any training input can capture the path. Thus, as long as the training process can cover common paths of a program, FarSight can generalize well to different inputs.

For all the applications used in our evaluation, our training time is around 30 minutes on a single A6000 GPU. The low training cost and the theoretical foundation of good input generalization make FarSight a viable solution for many on-prem data centers.

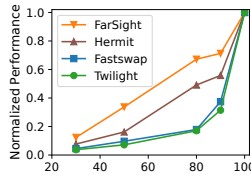

Figure 4: MCF performance.

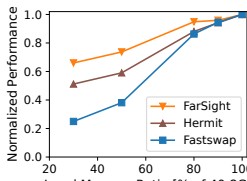

Figure 5: XGBoost performance.

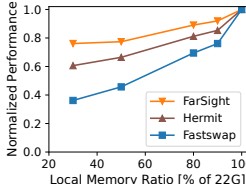

Figure 6: Page rank performance.

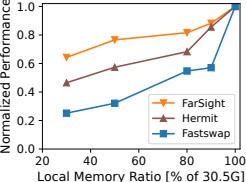

Figure 7: Shortest path performance.

## 4 EVALUATION RESULTS

**Implementation and environments.** We implemented FarSight with 5.5K lines of source code in the Linux kernel. We use a RetNet model architecture with 2240 parameters in 2 layers. We list our model hyperparameters in Appendix A.2. We evaluate FarSight on our private clusters. The compute node is running on a server equipped with a 28-core Intel Xeon Gold 5512U CPU (2.1 GHz) and 16 GB RAM. The memory node is running on a server equipped with a 16-core Intel Xeon Gold 5218 CPU (2.3 GHz) with 64 GB RAM. Both servers are connected with 100 Gbps Mellanox EDR-CX4 NIC through a 100Gbps RoCE ToR switch.

**Baselines.** We compare FarSight with three baselines: FastSwap (Amaro et al., 2020), Hermit (Qiao et al., 2023), and Twilight (Duong et al., 2024). FastSwap is a classic swap-based far-memory system implemented in the Linux kernel. Hermit is a SoTA far-memory system that improves FastSwap's swap-out performance. Both systems use default Linux prefetching, which only captures and prefetches sequential memory accesses. Twilight is a closed-source SoTA ML-based CPU prefetcher that has only shown results via simulation. As a CPU prefetcher, Twilight assumes visibility into every memory access—an assumption that does not hold in an OS-level far-memory system, where page tables obscure fine-grained access information. Therefore, we built a simulator based on the Twilight paper's algorithm, captured its prefetching trace using the simulator, and replayed the trace in the runtime far-memory system.

**Workloads.** We evaluate FarSight and baselines with four workloads, XGBoost (Chen & Guestrin, 2016), PageRank and shortest path (SSSP) in GAP benchmark suite (Beamer et al., 2017), and MCF from SPEC 2006 benchmark (Henning, 2006). XGBoost is a gradient boosted decision tree (GBDT) framework that trains an ensemble of trees, with each tree correcting errors defined by a loss function. We run the NYC Taxi dataset (Work & Donovan, 2016) on XGBoost for a classification task that consumes 40.5 GB of memory. The GAP Benchmark Suite is a collection of graph processing benchmarks designed to evaluate the performance of graph analytics systems. We Select Shortest Path (SSSP) and PageRank due to their prevalence in large-scale systems, with generated graphs of 4–16 million nodes and memory usage of 22–30.5 GB. MCF (Minimum Cost Flow) from SPEC 2006 Benchmark is a standard workload for CPU architecture evaluation. We use small input graphs ranging from 220MB to 390MB to evaluate MCF with Twilight as a baseline, because running Twilight on larger workloads takes weeks or longer.

**Model Training.** We trained each application's model on a smaller input that is different from the larger testing inputs. We trained MCF with the smallest graph provided by SPEC and tested it on a graph that is 3x larger. We trained PageRank and SSSP with graphs generated from the Gapbs-provided script, which are 10x smaller than the testing graphs. We trained XGBoost on a small dataset, "Adult Census Income", with around 50k samples; we tested it with the NYC Taxi dataset with 73 million samples.

### 4.1 END-TO-END APPLICATION PERFORMANCE

**Application performance.** Figures 4, 5, 6, and 7 present the end-to-end application performance of MCF, XGBoost, PageRank, and SSSP, respectively. For each set of experiments, we vary the application server's local memory size between 30% and 90% of the total application memory size (X axis) and measure the total application execution time (Y axis). For each result, we normalize the application execution time against that of running at full local-memory capacity, and higher Y-axis values are better.

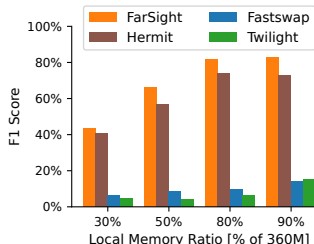
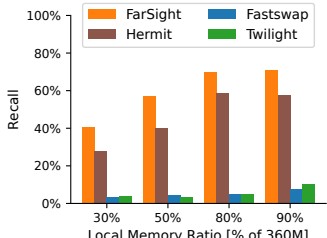
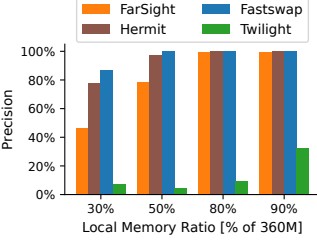

| Figure 8: MCF F1 score. | Figure 9: MCF recall. | Figure 10: MCF precision. |
|---|---|---|

FarSight consistently delivers superior performance across all four workloads, achieving up to 3.6 times and 2.6 times speedups over FastSwap and Hermit, and outperforming Twilight on MCF by up to 3.9 times. Comparing across local-memory size settings, FarSight achieves greater improvements when the local memory size is small—an environment especially challenging but useful for far-memory systems. A smaller local memory size increases the likelihood of missed accesses, amplifying the performance impact of an effective prefetching policy. Among the four workloads, FarSight performs best on MCF, where our DL-based prediction framework effectively captures the graph traversal pattern.

**Prefetch precision and recall.** To understand the effectiveness of FarSight's DL-based prefetching, we analyze the prefetch effectiveness by measuring precision, recall, and F1 score of its prediction and the baselines'. In the far-memory scenario, we define precision as the fraction of correctly predicted pages that are accessed by the application in local memory after prefetching (*i.e.*, true positive) over the total number of predictions (*i.e.*, prefetches). We define recall as the fraction of correctly predicted pages over the number of misses (page faults) plus the number of correctly predicted pages (*i.e.*, false negative + true positive). Figure 8, 9, and 10 plot the F1 score, recall, and precision of MCF workload, with other workloads' results in Appendix A.3.

Overall, FarSight's F1 score and recall are higher than all the baselines across settings and workloads, explaining its superior end-to-end application performance. FarSight eliminates 11% to 17% on-demand page faults compared to Hermit, 37% to 63% page faults compared to FastSwap and over 50% page faults compared to Twilight.

FarSight's precision is much higher than Twilight and on par with FastSwap and Hermit when local memory is large. When local memory is small, its precision is lower than FastSwap and Hermit. This is because FastSwap and Hermit rely on Linux's sequential prefetcher; only when a sequential pattern is detected do these systems perform prefetching, resulting in their high accuracy but low recall. FarSight takes a more agressive approach by trying to predict more complex access patterns, explaining its higher recall and relatively lower precision.

**Comparison to Twilight.** Twilight has the lowest precision and recall and performs the worst among all systems for several reasons. First, it involves two sub-predictions, one to determine the behavioral cluster an address belongs to and one to predict the subsequent for an address within a cluster. This process compounds inaccuracy in each sub-prediction, resulting in Twilight's overall lower accuracy. Second, it predicts the immediate next access. In far-memory systems, fetching a page requires one network round-trip. By the time the predicted page is fetched, the application usually already has the need to access it, still causing a page fault and explaining Twilight's low recall.

### 4.2 PERFORMANCE DEEP DIVE

**Generalization to different inputs.** To demonstrate FarSight's ability to generalize across inputs, we evaluate end-to-end application performance using multiple inputs of varying sizes. Figure 11 presents the results of MCF under 30% and 50% local memory ratios. The model is trained using a graph with 5K nodes and 50K edges and tested using three different graphs containing 20K to 40K nodes and over 200K edges. FarSight demonstrates strong generalization capability with one-time offline training, with an average performance improvement of $2.3\times$ over FastSwap across inputs. FarSight is able to generalize across inputs and adapt to larger inputs with models trained on smaller ones, thanks to our memory-access behavior and address layout decoupling.

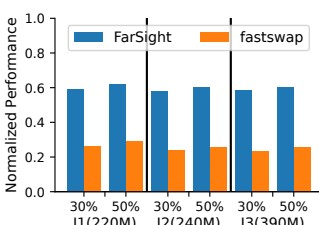

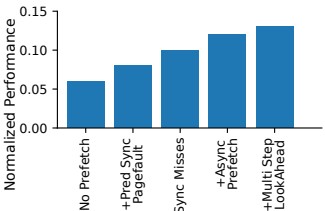

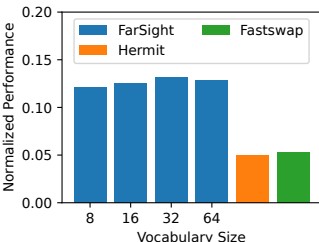

Figure 11: Handling input variance in MCF. *Three inputs used for MCF (220 MB, 240 MB, and 390 MB) tested on the same trained model.*

Figure 12: Performance breakdown of FarSight using MCF with 30% local memory. *Each bar adds one of FarSight's techniques at a time.*

Figure 13: Effect of vocabulary size on end to end performance.

**Ablation study.** To understand the performance benefits of each individual technique used in FarSight, we evaluate the system incrementally by adding one technique at a time. Figure 12 shows the breakdown with the 30% local memory configuration. We provide the ablation study of another setting in Appendix A.4. The leftmost bar represents a baseline configuration—the vanilla Linux setup without any prefetching. First, we add FarSight's model prediction but perform the prediction and prefetching synchronously on every page fault, which improves the baseline but not significantly. We then change the prediction and prefetching to only be triggered on page misses, as shown by the third bar. Although this gives FarSight less chance to perform prediction, we can significantly avoid the runtime performance overhead that would otherwise be incurred for prediction on page hits, resulting in improved performance.

So far, the prefetching FarSight performs is synchronous—application threads wait for it to finish. The fourth bar demonstrates the performance gain when performing prefetching asynchronously and allowing more flexible memory swapping, which improves the application performance further. The final technique incorporated into the full FarSight design incorporating the optimizations of look-ahead, batched prefetching. By predicting and issuing prefetches earlier, FarSight ensures that prefetched pages are more likely to arrive before they are actually needed.

**Sensitivity Tests** By default, FarSight uses a vocabulary and future-map size of 64 to strike a balance between memory overhead and prediction accuracy. To evaluate how FarSight performs under different vocabulary size, we vary $K$ between 8 and 64. As shown in Figure 13, FarSight demonstrates its robustness to various vocabulary size and consistently outperforms FastSwap and Hermit. FarSight's performance degrades slightly when $K$ is smaller than 32. This is because a small future map size cannot capture the possible outcomes of application memory accesses. For example, in an application with a recurring access pattern involving 16 frequently accessed pointers in an alternating sequence, a future map size of 8 entries will cause prediction pointers to overwrite each other, reducing prefetch accuracy and lowering the hit rate. Meanwhile, future map sizes larger than 64 add runtime performance overhead as they are less likely to be cached in CPU. Thus, we set the vocabulary size to be 64 by default, which works well for many typical applications.

## 5 CONCLUSION

We presented FarSight, a DL-based far-memory prefetching system with a core idea of decoupling the learning of application semantics from the runtime capturing of memory accesses. By doing so and with our set of optimization techniques, FarSight achieves overall application performance benefits over two recent far-memory systems, by up to 3.6 times and 2.6 times, and a SOTA CPU prefetcher by 3.9 times. FarSight demonstrates the feasibility of deploying modern ML techniques to solve performance-critical problems in complex runtime systems. Future researchers and practitioners could leverage lessons we learned and building blocks of FarSight's DL model, prediction problem presentation, and system-integration mechanisms.

## ETHICS STATEMENT

This material is based upon work supported by funding and gifts from multiple institutions. Any opinions, findings, conclusions, or recommendations expressed in this material are those of the authors and do not necessarily reflect the views of these institutions. Overall, the authors affirm that this work does not involve any ethical violations.

## REPRODUCIBILITY STATEMENT

We have taken steps to ensure that all experiments in this work are reproducible. The model architecture used in our experiments is fully described, and all model configuration and training parameters can be found in Appendix A.2. Hardware and training time requirements are detailed in Section 3.3. Regarding datasets and runtime environments, Section 4 provides the hardware used in our experiments, and all workloads and datasets referenced in the main text are publicly available. Finally, we provide an anonymous link to the source code in the supplementary materials to facilitate reproduction of our results.

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

# A APPENDIX

## A.1 USE OF LARGE LANGUAGE MODELS (LLMS)

We used LLM-based tools solely for polishing the writing, including improving grammar, phrasing, and readability. No part of the technical content, analysis, or experimental results was generated by an LLM. The authors remain fully responsible for the accuracy and integrity of the paper.

## A.2 MODEL PARAMETERS

The model's number of parameters and layers was chosen to balance inference speed and accuracy. FP16 operations on AVX-512 instruction sets are used for the attention layers, while quantized int8 operations are applied to the MLP layers. As summarized in Table 1, we employ standard multi-head attention (MHA) rather than grouped-query attention (GQA) (Ainslie et al., 2023), as our model is small and has only a few attention heads. Training hyper-parameters, particularly the learning rate, were selected following the guidelines in (Kaplan et al., 2020), adjusted for our model's parameter size.

Table 1: Model configuration and training hyper-parameters.

| Model Parameter | Value | Training Parameter | Value |
|---|---|---|---|
| Number of parameters | 2240 | Batch size ($B$) | 1024 |
| Hidden dimension ($d_{\text{model}}$) | 8 | Learning rate ($\eta$) | 0.003239 |
| Number of heads ($d_{\text{attn}}$) | 4 | Loss function | Cross Entropy |
| Number of layers ($n_{\text{layer}}$) | 2 | Optimizer | AdamW |
| Maximum sequence length ($T$) | 64 | | |
| Attention Mechanism | MHA | | |

## A.3 ADDITIONAL PRECISION AND RECALL RESULTS

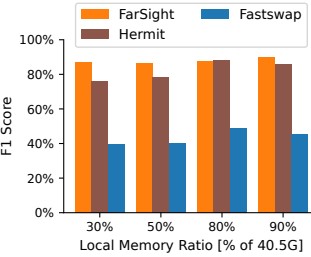 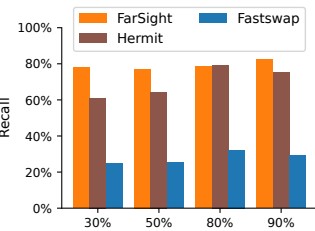 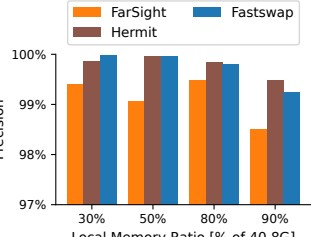

Figure 14: XGBoost F1 score        Figure 15: XGBoost recall        Figure 16: XGBoost precision

Figure 14, 15, 17, 18, 20, and 21 show additional F1 score and recall results of XGBoost, Page Rank and Shortest Path. As noted in the main text, Twilight is not included in the comparison for these three workloads because simulating memory traces and prefetching behaviors for memory footprints larger than 10GB would take months. FarSight shows a better F1 and recall score in majority of the data points. Compared to XGBoost and Shortest Path workloads, FarSight achieves higher recall and F1 scores, demonstrating a clear improvement. When compared to PageRank, FarSight achieves recall and F1 scores that are very to other baselines, reflecting the fact that FarSight performs well on this relatively simple memory access pattern. Despite the similarity in prediction metrics, FarSight still delivers better end-to-end performance thanks to its more effective prefetching mechanism, although the improvement margin is naturally smaller.

Regarding the precision results from Figure 16, 19, and 22, we observe that, consistent with the MCF results, precision is not the primary strength of FarSight, as Linux prefetch halts immediately when encountering non-sequential access patterns. Nevertheless, FarSight remains competitive across workloads.

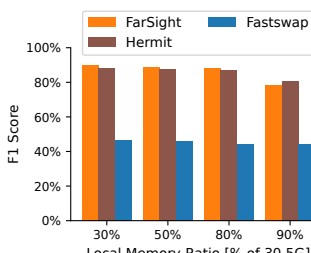
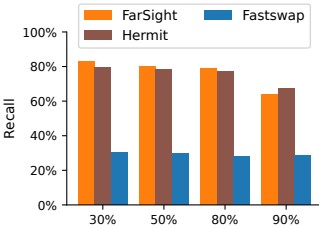
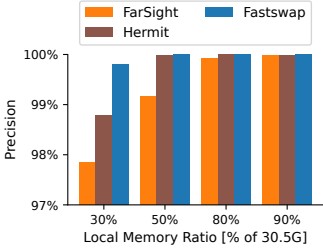

Figure 17: Shortest path F1 score

Figure 18: Shortest path recall

Figure 19: Shortest path precision

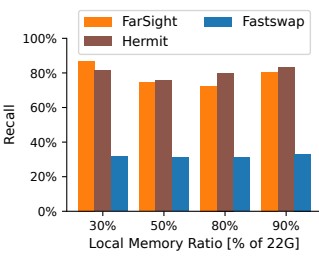
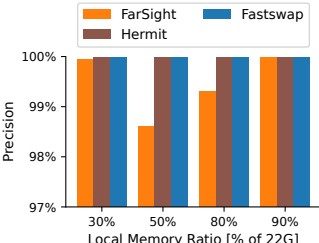

Figure 20: Page rank F1 score

Figure 21: Page rank recall

Figure 22: Page rank precision

## A.4 Additional ablation Results

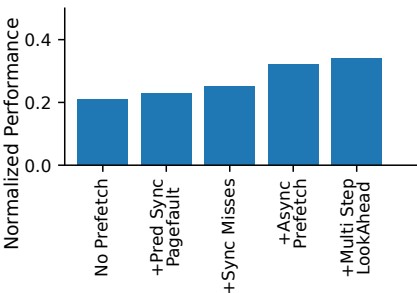

Figure 23: Performance breakdown of FarSight using MCF with $50\%$ local memory. *Each bar adds one of FarSight's techniques at a time.*

