# OpenReview forum: "Learning Semantics, Not Addresses: Runtime Neural Prefetching for Far Memory"
_ICLR.cc/2026/Conference — Submitted to ICLR 2026_

### Official Review · Reviewer_2VXJ · 2025-10-27

**Soundness:** 3
**Presentation:** 3
**Contribution:** 2
**Rating:** 4
**Confidence:** 3

**Summary:**

This paper proposes predicting the ordinal positions of future addresses rather than their exact memory addresses for remote memory prefetching. In addition, FarSight predicts page misses to reduce inference cost and uses a look-ahead distance to prefetch farther into the future, thereby hiding communication latency. Various experiments demonstrate its effectiveness.

**Strengths:**

- The paper is well structured and easy to follow.
- The proposed future map effectively reduces the vocabulary size of the deep learning model used for prediction.
- Experimental results show that FarSight achieves a high recall rate.

**Weaknesses:**

- The performance improvements on Shortest Path, PageRank, and XGBoost are limited compared with Hermit.
- The paper lacks an evaluation of end-to-end speedup. Low precision can lead to unnecessary prefetching, which might increase overall latency.
- The approach requires a warm-up stage to construct the future map. How long does this warm-up take?

**Questions:**

See weaknesses

---

### Official Review · Reviewer_Qc8D · 2025-10-31

**Soundness:** 3
**Presentation:** 3
**Contribution:** 1
**Rating:** 2
**Confidence:** 5

**Summary:**

Paper proposes a (small) neural network that can predict memory access patterns of programs. It can be used to predict addresses to be accessed in the future. Authors implemented this network inside of the Linux kernel and named it FarSight. The Linux-implementation prefetches addresses that are predicted by FarSight. Prefetching makes sure that the data is local (e.g., to CPU cache) when needed. The network is trained offline per program.  Importantly, during training, they present different inputs than the one for testing -- the testing instances are larger.

In summary, I do like the paper. However, the method is not practical, and the paper does not introduce models, ML, or even insights. As-is, I would prefer it in systems-like conferences, such that, researchers for those venues can perhaps make it more practical, draw insights from it, or perhaps introduce axioms for prefetching, such as invariance or equivariance on starting-address of data structures. At that point, the work could be more relevant to ICLR community, IMO. If other reviewers disagree with me, I could change my mind.

**Strengths:**

* Improving the speed of computation is good for almost everyone: faster compute = cheaper runtime = lower carbon footprint.
* Prefetching indeed speeds-up programs, e.g., especially in for-loops (I witnessed this first-hand)!
* Authors have implemented in Linux already!

**Weaknesses:**

* The main weakness -- which is not a real weakness -- is the relevance of this paper to the ICLR community. The paper is very system-like -- descriptive text but not a single equation or model. It would have been more relevant if it was presented in a way that appeals to the audience. For example, the actual model (mod, feeding into ML) should be written as equations -- they are less to process than lines of text, especially as we are used to see them on the regular.

I am not sure the method actually works, given the way it is described. Specifically, at the kernel level, the memory address space is shared among processes. Therefore:
1. starting a **static program** twice will almost never access the same memory locations. This would have been fine **if** the model is invariant (or equivariant) to the **starting index** of the data structures. Perhaps it is, but this is no where mentioned or proved in the paper.
2. Equally importantly, many memory addresses are contained within memory elements themselves (e.g., in the mentioned linked-list example). There are no guarantees that the elements are in any particular order. The paper does not address this fact.

The paper also does not compare with heuristic "traditional" baselines and only compares with methods developed in the last 4 years.

The outcome is not practical. Software changes all the time, due to changes in main code, library code, compiler versions, etc. It is unreasonable to keep updating FarSight models, one model per software, as the software changes -- in other-words, it is definitely not plug-and-play. One could argue that it is the right fit for particular workloads (e.g., matrix multiplications or pagerank, etc), and perhaps it is worth keeping it up-to-date for these routines. However, it is probably much easier to just amend "prefetch" instructions within the C++ code which will be better than any learned model -- I do insert prefetch instructions especially in for-loops when operating under resource constraints.

**Questions:**

While being a very systems paper (to the point it is outside the scope of ICLR), the paper misses important system details.

1. How will FarSight run on a system that has many concurrent software?
2. Will it just activate for the one it trained on?
3. How is FarSight trained (offline)?
4. Using tensorflow/jax/pytorch, or did the authors implement backprop within the 5.5K lines? If it is the first, then how are the parameters saved and then loaded by the 5.5K code lines?

---

### Official Review · Reviewer_LWiy · 2025-10-31

**Soundness:** 2
**Presentation:** 3
**Contribution:** 3
**Rating:** 4
**Confidence:** 5

**Summary:**

This paper presents a deep learning-based prefetching system labeled FarSight for far-memory environments implemented in the Linux kernel. The core innovation decouples application semantics from runtime memory addresses by training a compact RetNet model (2240 parameters) to predict ordinal indices (0 to K-1) rather than actual memory addresses. These ordinals are resolved at runtime through "future maps" that maintain mappings from indices to actual page addresses. The system achieves up to 3.6x speedup over FastSwap, 2.6x over Hermit, and 3.9x over Twilight across four workloads while maintaining <600ns prediction latency.

**Strengths:**

• The ordinal vocabulary representation reduces the intractable address space problem (2^48 addresses) to a learnable vocabulary of K=64. This enables accurate prediction with a model that fits in L1/L2 cache
    • This appears to the first Linux-based far-memory system to successfully deploy deep learning in production rather than simulation-only results, with 5.5K lines of kernel code demonstrating real-world feasibility
    • Multiple clever optimizations work synergistically: using page misses rather than all memory accesses (observable via page faults without overhead), incorporating program counter context, rotary positional encoding to reuse cached computations, and look-ahead prefetching to cover network latency
    • Strong generalization demonstrated in Figure 11 where a model trained on 5K-node graphs successfully handles 40K-node graphs (8x larger), with 2.3x average speedup, validating the semantic learning claim
    • Comprehensive ablation study (Figure 12) clearly demonstrates the contribution of each technique, showing progression from 0.02 to 0.15 normalized performance as components are added
    • Addresses all three critical challenges simultaneously (latency, accuracy, generalization) that prevented prior ML-based prefetchers from practical deployment
• The literature review and bibliography list is sufficient and satisfactory.

**Weaknesses:**

A) Empirical evaluation weaknesses (Please also see questions for how to improve):
i) Lack of comprehensive evaluation, specifically paucity of **diverse** workloads; they evaluate only four workloads, with three being graph-based (MCF, PageRank, SSSP). Paper claims applicability to "graph processing, tree and index structures, pointer chasing, and recursive data structures" (lines 37-40) but provides no evidence beyond graph traversals, which is only one of many challenging graph analytic metrics. Missing databases (despite mentioning B-trees), key-value stores, and scientific computing workloads. Suggestion - Need 3-4 additional workloads from **different domains** to support their claims of broad applicability of FarSight.

ii) Paper mentions both Twilight and T-LITE but only evaluates against Twilight, ignoring T-LITE entirely. Twilight comparison itself is problematic: reimplemented via simulation (ChampSim?) rather than using original code, only compared on smallest workload (MCF 220-390MB), excluded from main workloads because "simulating takes weeks." This is almost contradictory i.e., if Twilight is too heavy to evaluate, T-LITE (the lightweight version) should be the natural comparison. Not providing any explanation for excluding a T-LITE comparison makes baseline comparison appear incomplete or cherry-picked.

iii) K=64 vocabulary size lacks justification—sensitivity analysis (Figure 13) only tests up to K=64, leaving unknown whether K=128/256 would improve performance. No analysis of what fraction of pages need >64 successors in real workloads. FarSight has notably lower precision than baselines at 30% local memory (Figures 10, 16, 19, 22), dismissed as "more aggressive" without analyzing misprediction costs. No breakdown of where/why predictions fail or which code patterns confuse the model.

Theoretical weaknesses:

-The paper is missing key algorithmic design and analysis details for e.g., what is the strategy for future map initialization, what does the system do at boundary events e.g., handling when K entries are filled, what is the collision rate for (address mod K, PC mod K) pairs and how does it influence the design? There is no pseudocode. The training methodology tests only one input per workload with no sensitivity analysis despite generalization being a key claim.

**Questions:**

Q1: Can you provide results on 2-3 additional non-graph workloads (database, KV store, scientific computing)? This would substantially strengthen generalization claims.
Q2: Why was T-LITE not evaluated despite being mentioned in line 119? If Twilight is too expensive, isn't T-LITE (the lightweight version) the appropriate comparison?
Q3: What is the empirical distribution of successor counts per page? How many pages have >64 successors and what fraction of accesses do they represent?
Q4: Can you provide breakdown of prediction failures (never-accessed vs. wrong ordinals vs. timing issues) and analysis of which code patterns cause systematic failures?
Q5: Have you tested K > 64? Where does performance plateau?
Q6: Can you validate training robustness by testing at least one workload with 2-3 different training inputs?

---

### Official Review · Reviewer_Nh4R · 2025-10-31

**Soundness:** 2
**Presentation:** 3
**Contribution:** 1
**Rating:** 2
**Confidence:** 4

**Summary:**

The paper proposes FarSight, a Linux‑kernel prefetching framework for far‑memory systems that learns application semantics offline and resolves actual addresses online. The key idea is to decouple prediction from addresses: a small DL model predicts, over a fixed ordinal vocabulary, which relationship/next outcome is likely after a page miss. At runtime a future map per page translates the predicted ordinal to a concrete virtual page to prefetch. The model is a very small Retentive Network (RetNet) with around 2.2K parameters, augmented by (i) rotary position encoding reuse so history windows share cached context, (ii) look‑ahead prediction to hide RDMA latency, and (iii) batched, asynchronous prefetch. The system is implemented as 5.5K LoC in the Linux swap path. On four workloads (SPEC2006 MCF, GAP PageRank/SSSP, XGBoost on NYC Taxi), FarSight reports up to $3.6 \times$ speedup over FastSwap, $2.6 \times$ over Hermit, and $3.9 \times$ over a re‑implemented simulator of Twilight (NN CPU cache prefetcher) on MCF; precision/recall/F1 analyses and ablations on vocabulary size and components are provided.

**Strengths:**

The semantics-addresses decoupling via an ordinal vocabulary and per‑page future maps is a neat formulation that squarely addresses the problem of the 64‑bit address space for sequence models. This differs from prior ML prefetchers that predict concrete addresses or small fixed offsets. Applying a very small RetNet with constant‑time inference to OS‑level far‑memory prefetching, plus the rotary‑reuse encoding to avoid recomputation across sliding windows, is a pragmatic systems‑ML combination. The problem setup and constraints for far memory are well‑articulated. The method is explained with concrete examples (e.g., Fig. 2) and a clear workflow diagram (Fig. 1).

**Weaknesses:**

The paper compares against FastSwap and Hermit (Linux‑prefetch based) and against a home‑built simulator of Twilight only on MCF, replayed into the far‑memory runtime (Sec. 4, "Baselines"). This raises fairness and validity questions: the mapping from Twilight’s cache‑line predictions to page‑granular far‑memory prefetch is nontrivial and may handicap it; details of the simulator (e.g., clustering choices, history length, throttling, look‑ahead) are sparse. Please provide a more thorough description of the Twilight re‑implementation, and, if possible, add a simple address‑agnostic ML baseline (e.g., n‑gram/Markov over future‑map ordinals) to disentangle the value of the RetNet choice from the decoupling idea itself.

Figures 5-7 normalize to the "full local‑memory capacity" run, but the compute node is reported to have 16 GB RAM while datasets reach 22-40.5 GB (Sec. 4 "Implementation and environments"). It is unclear how the 100% local baseline was obtained for PageRank/SSSP/XGBoost (same machine with larger DRAM? a different host? memory compression?). Please clarify the machine used for the 100% local baselines and ensure the environment is otherwise identical (CPU/NIC/kernel).

The paper cites <600 ns per prediction and ~20 KB weights/metadata locality (Sec. 3.2), but there is no end‑to‑end CPU utilization or instruction‑per‑fault accounting, no NIC bandwidth overhead breakdown due to wrong prefetches, and no memory‑footprint numbers for the future maps. With K=64, a naive per‑page future map (64 pointers) could be substantial if maintained for many pages. Please report: (a) predictor CPU time/IPC per page fault; (b) additional RDMA bandwidth and its share of link capacity; (c) memory overhead per future map and total footprint vs. number of tracked pages; (d) amortized cost under multiprogramming.

Figure 1 distinguishes "Full Future Maps" and "Cached Future Maps", but the text is ambiguous about where/how many maps are kept, eviction policy, and how maps survive page eviction or process restarts (Sec. 3.1 & Fig. 1). Spell out the data structure, indexing (per‑page? per‑region?), replacement, and whether maps are persisted, regenerated, or pre‑warmed.

Inputs use PC % K and (page address) % K (Sec. 3.1). While history context helps, this introduces aliasing. The paper lacks an analysis of collision rates and their impact on accuracy across workloads, nor an ablation replacing modulo with a learned hash or larger embeddings. Please add an evaluation of aliasing (e.g., confusion matrices over ordinals; impact of larger K values and/or simple learned embeddings).

**Questions:**

How many future maps are live at a time for the large workloads, and what is their exact per‑map size? Please report total memory overhead (MB/GB) devoted to maps at each local‑memory ratio.

Where were the “full local” runs executed for PageRank/SSSP/XGBoost, given the 16 GB compute node? Were CPU/NIC/kernel identical?

Provide implementation details, including hyper‑parameters, how you translated cache‑line prefetches to pages, and whether look‑ahead or asynchronous issuing were enabled for Twilight to make the comparison fair.

What fraction of time does the history window map to non‑unique token sequences because of modulo? Any evidence of error spikes tied to collisions?

What is the per‑fault prediction latency distribution (not just mean) and the CPU utilization of the background predictor at high fault rates? Can the L1/L2 residency guarantees always hold under heavy multiprogramming?

Show sensitivity to network jitter and bursty phases; what are the fallbacks when prefetched pages arrive late (do you cancel/age out queued prefetches)?

---

### Author Response · Authors · 2025-11-15

We thank the reviewers for their thoughtful and constructive feedback. We acknowledge the concerns raised and appreciate the valuable suggestions. We will incorporate the reviewers’ advice and improve the work for future submissions.

---

### Meta-Review · Area_Chair_Vh8o · 2026-01-06

**Summary:**

The reviewers acknowledge that the paper presents an interesting systems contribution in the form of FarSight, a Linux-kernel–level far-memory prefetching framework that decouples application semantics from concrete memory addresses and demonstrates substantial speedups over prior far-memory systems on several workloads. The idea of predicting ordinal access patterns with a compact neural model and resolving them at runtime is viewed as technically sound and thoughtfully engineered. However, multiple reviewers raised concerns about the relevance and positioning of the work for the ICLR audience, the limited scope and diversity of the evaluation, and insufficient transparency and fairness in baseline comparisons and system-level overhead analysis.

**Reviewer Concerns:**

Although the rebuttal acknowledged the feedback, key concerns remain unresolved, including incomplete baseline coverage and fairness, limited workload diversity, missing end-to-end overhead and resource accounting, unclear scalability and robustness under realistic multiprogramming and address-space variability, and the lack of broader machine learning insights beyond a well-engineered systems implementation.

**Reviewer Scores:**

Reviewer scores would likely remain unchanged after discussion.

---

### Decision · Program_Chairs · 2026-01-26

Reject